# Green Fruit Detection with a Small Dataset under a Similar Color Background Based on the Improved YOLOv5-AT

**DOI:** 10.3390/foods13071060

**Published:** 2024-03-29

**Authors:** Xinglan Fu, Shilin Zhao, Chenghao Wang, Xuhong Tang, Dan Tao, Guanglin Li, Leizi Jiao, Daming Dong

**Affiliations:** 1College of Engineering and Technology, Southwest University, Chongqing 400716, China; fxl2021@swu.edu.cn (X.F.); 13273627393@163.com (S.Z.); wangchh2003@163.com (C.W.); 15979352499@163.com (X.T.); liguanglin@swu.edu.cn (G.L.); 2College of Electrical and Automation Engineering, East China Jiaotong University, Nanchang 330013, China; 3090@ecjtu.edu.cn; 3Research Center of Intelligent Equipment, Beijing Academy of Agriculture and Forestry Sciences, Beijing 100097, China; jiaolz@nercita.org.cn

**Keywords:** green fruit, precision detection, feature information, data augmentation methods, channel attention module, spatial attention module

## Abstract

Green fruit detection is of great significance for estimating orchard yield and the allocation of water and fertilizer. However, due to the similar colors of green fruit and the background of images, the complexity of backgrounds and the difficulty in collecting green fruit datasets, there is currently no accurate and convenient green fruit detection method available for small datasets. The YOLO object detection model, a representative of the single-stage detection framework, has the advantages of a flexible structure, fast inference speed and excellent versatility. In this study, we proposed a model based on the improved YOLOv5 model that combined data augmentation methods to detect green fruit in a small dataset with a background of similar color. In the improved YOLOv5 model (YOLOv5-AT), a Conv-AT block and SA and CA blocks were designed to construct feature information from different perspectives and improve the accuracy by conveying local key information to the deeper layer. The proposed method was applied to green oranges, green tomatoes and green persimmons, and the *mAPs* were higher than those of other YOLO object detection models, reaching 84.6%, 98.0% and 85.1%, respectively. Furthermore, taking green oranges as an example, a *mAP* of 82.2% was obtained on the basis of retaining 50% of the original dataset (163 images), which was only 2.4% lower than that obtained when using 100% of the dataset (326 images) for training. Thus, the YOLOv5-AT model combined with data augmentation methods can effectively achieve accurate detection in small green fruit datasets under a similar color background. These research results could provide supportive data for improving the efficiency of agricultural production.

## 1. Introduction

Fruits are a vital part of the human diet due to their nutritional properties, vitamin content, dietary fiber, phytochemicals and antioxidants, and fruit production is an important part of the global agricultural sector. However, most producers focus solely on the detection of ripe fruits and ignore the importance of green fruit detection in traditional agricultural management processes. Detecting green fruit can not only allow for early orchard yield estimation and optimizing and guiding the allocation of water and fertilizer but can also provide a basis for intelligent picking, which could improve agricultural production efficiency. Therefore, the early detection of green fruits has practical significance for agricultural production.

Detecting green fruit is made more difficult by the fact that the detection targets have a similar color to the background. Therefore, in early studies, researchers proposed the use of RGB images of fruits combined with image depth information, thermal image information or multispectral image information to detect green fruit under similar color backgrounds. For example, Gan et al. [1] proposed a new color thermal combination probability algorithm, which fused color and thermal image information and obtained 90.4% precision in the detection of green oranges. Liu et al. [2] used image depth information combined with RGB image information to detect oranges with different occlusion levels, achieving an accuracy of 79.17% on average. Although the above studies achieved the detection of green fruits under backgrounds with a similar color and resolved the poor detection accuracy caused by complex backgrounds, there was variable light and overlap and occlusion between the detection target and other plant parts (Tang et al. [3]). Some problems persisted, such as the poor versatility of the model; expensive equipment (thermal imager, RGB-D camera, etc.); complex operation and influence by temperature and weather, meaning that it is difficult to use this method to satisfy the actual production and application requirements.

In recent years, with the rapid development of computer technology, machine vision combined with deep learning has been widely used in autonomous vehicles (Feng et al. [4]), surveillance (Gupta et al. [5]), medical imaging (Varoquaux et al. [6]), agriculture (Kamilaris et al. [7]) and other fields (Bourilkov et al. [8]). Therefore, many researchers have tried to apply this technology to achieve the high-precision detection of fruits (Koirala et al. [9]; Gongal et al. [10]). For example, Parvathi et al. [11] used Faster-R-CNN to identify the maturity stages of coconuts (2000 images used for the model) and achieved a *mAP* of 89.4%. Villacrés et al. [12] used Faster R-CNN to identify cherries (15,000 images used for the model) and achieved an accuracy of 85%. Yu et al. [13] proposed an object detection algorithm based on Mask-RCNN, which was applied to strawberry detection (2000 images used for the model), and the precision reached about 95.78%. Liang et al. [14] used a SSD algorithm to detect on-tree mangos (1900 images used for the model) and achieved a *F*1 of 91.1%. Although the above studies achieved high levels of accuracy and good generalization for fruit detection, they all required a large number of datasets for modeling. However, in practical applications, it is difficult to collect data under different weather and light conditions, as this requires a lot of manpower, material and financial resources. Additionally, it is often difficult to obtain sufficient and high-quality datasets, whereas the low quality of the small datasets used in object detection algorithms can lead to problems such as data imbalance, overfitting and direct reductions in the generalization ability and performance of the model. Therefore, a fast and efficient small dataset (Shaikhina et al. [15]) training model was developed to achieve the accurate detection of green fruits.

Object detection algorithms are mainly divided into two categories: two-stage object detection algorithms, including RCNN, Fast-RCNN, Mask-RCNN and so on, and single-stage object detection algorithms, such as the SSD algorithm and YOLO series algorithm. The YOLO series algorithm, as a representative framework that derives its single-stage detection from YOLOv1-YOLOv8 (Terven et al. [16]), has the advantages of a more flexible structure, faster inference speed and stronger versatility compared to other object detection models (Jiang et al. [17]). YOLOv3 (Redmon et al. [18]) accelerates the application of object detection by introducing multi-scale prediction and the backbone, neck and head network architecture and loss function improvement. On the basis of YOLOv4, YOLOv5 further simplified the structure, making the model smaller and more flexible and the image inference speed faster and closer to natural production and life. YOLOv7 (Wang et al. [19]) uses an extended efficient layer aggregation (E-ELAN) module and a concatenation-based architecture to improve its accuracy without compromising the inference speed. In addition to the above iterative updates, YOLOR, YOLOX, DAMO-YOLO, PP-YOLO, etc., a variety of models based on the core idea of the YOLO models were proposed for different object detection tasks. For example, Salman et al. [20] designed an automatic grading diagnosis system for prostate cancer based on the YOLO algorithm (500 images used for the model) and achieved an accuracy of 89–97%. Jiang et al. [21] designed an improved YOLOv4 model to detect ships (1160 images used for the model) and achieved an accuracy of 90.37%, which showed that YOLO can be widely used in various fields. In the agricultural field, Wu et al. [22] designed the improved YOLOv5-s model to detect sugarcane seedlings (988 images used for the model), achieving a *mAP* of 93.1%, and the improved algorithm was applied to an actual production robot. Zheng et al. [23] proposed the PrunedYOLO-Tracker algorithm for cow behavior recognition and tracking (400 videos used for the model), which achieved a *mAP* of 87.7%. Liu et al. [24] proposed the AFF-YOLOX algorithm to detect the hatching information of multiple duck eggs on a standard hatching plate (2111 images used for the model), which achieved a *mAP* of 97.55%. In addition, Zhang et al. [25] proposed SwinT-YOLO to detect the position and size of corn ears (1000 images used for the model), which achieved an accuracy of 95.11%. It can be seen that the YOLO model possesses both good accuracy and a wide application range in the field of object detection.

As demonstrated above, we proposed a green fruit detection model based on an improved YOLOv5 to address the issue of the low accuracy of small datasets and to detect green fruit targets under a similar background color. The main contributions of this study are as follows: Firstly, the Conv-AT block is designed to extract and preserve important features from multiple perspectives. Secondly, the Conv-AT block incorporates SA and CA mechanisms to capture the essential spatial and channel dimensions of input features. Thirdly, data augmentation methods such as SAM Copy-Paste are applied to expand the dataset, mitigating the problems of small datasets.

## 2. Materials and Methods

### 2.1. Acquisition of Images

The images were collected in various orchards in Chongqing, China, from 9:00 to 17:00 in sunny, cloudy and overcast weather conditions. The image acquisition equipment was a Panasonic DMC_LX5GK digital camera with a resolution of 2560 × 1920. A total of 468 green orange images (the Citrus Research Institute of Southwest University), 281 green persimmon images (Xishanping, Beibei District, Chongqing Agricultural Park) and 313 green tomato images (Luozhi Town, Yubei District, Chongqing) were collected (Figure 1). When modeling, the training set and test set were divided according to a ratio of 7:3, as shown in Table 1.

### 2.2. The Algorithm Principle of YOLOv5-AT

In this study, the YOLOv5-AT model is proposed based on the YOLOv5 model, and its structure is shown in Figure 2. The original code of the YOLOv5-AT model is provided in the Appendix A.

The input module randomly implements SAM Copy-Paste, Mosaic, Mixup and other basic image data augmentation methods according to different probabilities to expand the data of the original small dataset (Shorten et al. [26]). In the input module, the calculation of the adaptive anchor box and the adaptive scaling of the image are realized at the same time to obtain the optimal anchor box size and images adapted to the target size.

The backbone network is mainly composed of a CBS block, Conv-AT block, C3 block and SPPF block. The CBS block is used to reintegrate and extract the input features. The Conv-AT block is an attention integration and feature extraction module based on the attention mechanism and concatenation operation, which is introduced to enhance the learning ability of the model, eliminate the computational bottleneck and accelerate the convergence speed of the network. The C3 block (Park et al. [27]) divides the input feature map into two parts and merges them through a hierarchical structure across stages to reduce the computational cost and ensure detection accuracy. The SPPF block performs maximum pooling using kernels of different sizes. Then, a concatenation operation is performed to fuse feature maps of different scales (Li et al. [28]).

The neck network designs a Feature Pyramid Network (FPN) and pixel aggregation network (PAN) to achieve multi-scale feature fusion. The FPN fuses the lower layer feature maps by down-sampling the upper-layer feature maps through the convolutional layer and conveys the deeper semantic information. The PAN conveys the localization features from the bottom feature map to the top feature map through up-sampling (Lin et al. [29]). The FPN and the PAN adequately fuse the multi-scale features extracted from the backbone network and enhance the feature fusion ability of the neck network.

The prediction head mainly uses the feature information from the neck network for multi-scale object detection and outputs three feature maps of different sizes to detect small, medium and large objects, respectively (Ma et al. [30]). Each output generates a vector of information related to the target object, including its the probability, score and bounding box location.

#### 2.2.1. Conv-AT Block

In the backbone of YOLO-AT, the Conv-AT block is used to better extract the feature information obtained through the CSP block based on the attention mechanism, in which the traditional adding operation was abandoned and the concatenation operation was adopted as the main means of transferring the feature information, which can better retain the structure information of the input features. For the YOLO series object detection models, the proper retention of feature information from the perspectives of space and channel has a certain role in improving the accuracy of the network. Furthermore, the CBR block is designed as the basic component of the Conv-AT block to extract features. The ReLU activation function is selected in the CBR block to operate the input features in a nonlinear manner compared to the CBS block. The ReLU activation function can accelerate the convergence speed of the model and prevent gradient disappearance or gradient explosion compared to the SiLU activation function (Schmidt-Hieber et al. [31]). Further, the channel attention (CA) and spatial attention (SA) mechanisms were designed to obtain feature information from the two perspectives of channel and space, so as to improve the detection precision of the model. Finally, we added the BN operation again at the end of the Conv-AT block in order to prevent the gradient explosion caused by the concatenation operation, which also speeds up the convergence of the model.

*X^C^*^×*H*×*W*^ is the input feature, with *C*, *H* and *W* representing the number of channels, height and width of the input features, respectively. *P*(*X*) is the first feature map to be concatenated. *P*(*X*) can be expressed as
(1)$$P(X)\;=\mathrm{concat}\left\{F_{\mathrm{CA}}\left(X\right);\;F_{\mathrm{CBR}}\left(X\right)\right\}$$
where $F_{\mathrm{CBR}}\left(X\right)$ denotes the CBR operation on the input feature, and $F_{\mathrm{CA}}\left(X\right)$ denotes the CA operation on the input feature.

Then, *P*(*X*) is concatenated with the input feature map that is operated by the SA mechanism to obtain H(X). *H*(*X*) can be expressed as
(2)$$H(X)=\mathrm{concat}\left\{F_{\mathrm{CBR}}\left(P(X)\right);\;F_{\mathrm{SA}}\left(X\right)\right\}$$
where *F*_SA_(*X*) denotes the SA operation on the input feature.

Next, *H*(*X*) is concatenated with the input feature map that is operated by the CBR block to obtain *W*(*X*). *W*(*X*) can be expressed as
(3)$$W(X)\;=\mathrm{concat}\left\{F_{\mathrm{CBR}}\left(H(X)\right);\;F_{\mathrm{CBR}}\left(X\right)\right\}$$

Finally, the output *Y*(*X*) can be expressed as
(4)$$Y(X)=\mathrm{BN}(F_{\mathrm{CBR}}\left(W\left(X\right)\right))\;$$
where BN(*X*) denotes the batch normalization operation.

#### 2.2.2. CA and SA Mechanisms of the Conv-AT Block

In order to extract better feature information from the perspectives of space and channels, the CA and SA blocks were designed in the Conv-AT block, and their structures are shown in Figure 3.

Figure 3a shows the structure of the CA block: a lightweight attention mechanism that not only captures cross-channel information but also considers orientation-aware and location-sensitive information (Woo et al. [32]). CA is achieved using coordinate information embedding and coordinate attention generation. The calculation formula is shown in (5):(5)$$F_{\mathrm{CA}}\left(X\right)=X\times \sigma \left(\mathrm{MLP}\left(\mathrm{Avgpool}\left(X\right)\right)+\mathrm{MLP}\left(\mathrm{Maxpool}\left(X\right)\right)\right)$$
where $F_{\mathrm{CA}}(X)$ is the channel attention operation on the input feature *X*, and Avgpool(X) and Maxpool(X) are the average and max pooling operations. MLP represents the multi-layer perception layer, and σ is the sigmoid activation function.

Figure 3b shows the structure of the SA block, which applies the average pooling and maximum pooling to aggregate channel information on the channel perspective. The obtained feature information is conveyed to the MLP for processing, and then, the weight of the spatial angle feature information is obtained using the sigmoid activation function. The calculation formula is shown in (6):(6)$$\;F_{\mathrm{SA}}\left(X\right)=X\times \sigma \left(\mathrm{concat}\{\mathrm{MLP}\left(\mathrm{Avgpool}\left(X\right)\right);\mathrm{MLP}\left(\mathrm{Maxpool}\left(X\right)\right)\}\right)$$
where $F_{\mathrm{SA}}(X)$ is the spatial attention operation on the input feature *X*.

#### 2.2.3. Data Augmentation Methods


(1)SAM Copy-Paste


Copy-Paste is a data augmentation method proposed by Ghiasi et al. [33] for semantic segmentation models, where the core objective is to copy instances from the original image and paste them into another image according to the annotated contours of the instances. This method can effectively improve the diversity of a dataset and expand the number of training sets. However, the original Copy-Paste data augmentation method has the disadvantages of difficult contour annotation, high annotation cost and large time consumption (10 min per image). The Segment Anything Model (SAM) (Kirillov et al. [34]) can accurately obtain all the contour information in an image, which can replace manual annotation and improve efficiency. It only takes 10 s to complete the segmentation of a single image. Therefore, the combination of Copy-Paste and SAM is proposed for the first time in this study. Firstly, the SAM model was used to accurately obtain the contour information of fruits and leaves. Then, after geometric transformation of the obtained contour information, a random part of the information was selected. Finally, the selected contour information was combined with Copy-Paste to expand the dataset. By pasting leaves and green fruits, the model can better simulate various complex real conditions of green fruits and increase the diversity of the dataset (Zhang et al. [35]). The SAM Copy-Paste model was applied with a probability of 1 and pasted 4 instances of green oranges and 15 instances of leaves into each image. Its workflow diagram is shown in Figure 4.


(2)Mosaic


The Mosaic method (Bochkovskiy et al. [36]) involves taking a random number of images (usually four), scaling the width and height of each image to a preset size and splicing the Mosaic image in a random order according to the set center point of the image. Then, the border is deleted or filled. Finally, basic data augmentation transformation (rotating, flipping, cropping and so on) is performed on the generated Mosaic image to obtain a new image. The Mosaic data augmentation method can significantly increase the diversity of the original data and enhance the robustness of the model. The Mosaic method was applied with a probability of 0.8. Its workflow diagram is shown in Figure 5.


(3)Mixup


The Mixup method (Zhang et al. [37]) involves randomly selecting two training samples and using linear interpolation to generate a new training sample and the corresponding label. The calculation formulas are shown in (7) and (8):(7)$$\tilde{x}=\lambda x_{i}+\left(1-\lambda \right)x_{j}$$
(8)$$\tilde{y}=\mathrm{concat}\{y_{i}{;y}_{j}\}$$
where $x_{i}$ and $x_{j}$ are two random input images, λ is the probability value obeying Beta distribution with the parameter α and $y_{i}$ and $y_{j}$ are the labels corresponding to $x_{i}$ and $x_{j}$. Mixup was applied with a probability of 0.3. Its workflow diagram is shown in Figure 6.

### 2.3. Evaluation Criteria

In order to evaluate the performance of the YOLOv5-AT model through the detection results, precision (*P*), recall (*R*), average precision (*AP*) and *F*1 score (*F*1) were used as the evaluation indicators. *P* represents the proportion of true positive samples in the positive samples predicted by the model. *R* represents the proportion of positive samples predicted by the detector in the total positive samples (Hsu et al. [38]). However, *P* and *R* fail to evaluate detection accuracy directly. Then, *AP* and *F*1 are introduced to evaluate the capability of the detection network. *AP* represents the average precision rate in the detection. *F*1 is the harmonic mean of precision and recall. A higher *AP* and *F*1 index denote the higher accuracy of the detection model. The calculation formulas are shown in (9)–(13):(9)$$\;P=\frac{\mathrm{TP}}{\mathrm{TP}+\mathrm{FP}}$$
(10)$$R=\frac{\mathrm{TP}}{\mathrm{TP}+\mathrm{FN}}$$
(11)$$\mathrm{AP}=\int_{0}^{1}P(R)dR$$
(12)$$\mathrm{mAP}=\frac{\sum_{i=1}^{n}AP_{i}}{n}$$
(13)$$F1=\frac{2\times P\times R}{P+R}$$
where *TP* denotes the number of positive samples predicted as positive, *FN* denotes the number of negative samples predicted as negative, *FP* denotes the number of negative samples predicted as positive and *n* is the number of categories. The intersection set *IoU* indicates the overlap ratio between the predicted bounding box and true bounding box. In this research, the threshold for the *IoU* was set as 0.5. The sample was defined as a *TP* when the *IoU* was greater than 0.5. If the *IoU* was less than 0.5, this sample was defined as a false sample or *FP*.

## 3. Results and Discussion

### 3.1. Comparison of Different YOLO Models

The performance of our YOLOv5-AT model was evaluated in comparison with existing YOLO models, including YOLOv5, YOLOv3, YOLOv3-AT, YOLOv7 and YOLOv7-AT, using the same green orange dataset. Three data augmentation methods (SAM Copy-Paste, Mosaic and Mixup) were applied simultaneously. The comparison results are shown in Table 2 and Figure 7.

According to Table 2, the *mAPs* of the YOLOv3, YOLOv5 and YOLOv7 models rose by 0.7%, 1.7% and 2.4%, respectively, after applying the Conv-AT block, and the *P*, *R* and *F*1 values were also improved. It can be seen that the Conv-AT block improved the accuracy of the model. Compared to the *mAP* values of YOLOv3-AT (84.3%) and YOLOv7-AT (79.1%), the YOLOv5-AT model had the best detection ability and showed a stable *mAP* at 84.6%. The YOLOv5-AT showed improvement in the *P*, *R* and *F*1 score of 5.3%, 1.8% and 3.4%, respectively, compared to the YOLOv7-AT model. Simultaneously, the YOLOv5-AT model showed improvements of 1.2%, 1.1% and 0.7% in the *P*, *R* and *F*1 score, respectively, compared to the YOLOv3-AT model.

According to Figure 7, compared to the benchmark model, YOLOv3-AT and YOLOv5-AT detected the targets that the benchmark model failed to detect. The accuracy of the benchmark model further improved after applying the Conv-AT block. For the YOLOv7 model, after applying the Conv-AT block, the confidence in its detection of green oranges also improved. The above experimental results show that the YOLOv5-AT model had the highest detection accuracy for green oranges under a similar background color compared to the other models.

### 3.2. Comparison of Different Data Augmentation Methods

In this study, different data augmentation methods, including All-Augment (SAM Copy-Paste, Mosaic and Mixup and so on); No-Augment (no data augmentation method applied) and Mosaic, Mixup and SAM Copy-Paste (only a single data augmentation method applied), were applied to the YOLOv5-AT model. The changes in the *mAP* values of the training process and precision recall curve are shown in Figure 8.

Figure 8a shows that, in the YOLOv5-AT model, the model reached the convergence stage at 20–40 iterations, and different data enhancement methods had different effects on the model. The Mosaic method allowed the YOLO model to achieve the highest *mAP* in the convergence stage when using every data augmentation method alone. The SAM Copy-Paste method showed the best reduction in *mAP* fluctuation. The Mixup method improved the *mAP* of the model but caused fluctuations. The SAM Copy-Paste suppressed the fluctuations caused by Mixup when the three data augmentation methods were applied to the model at the same time. Therefore, the combination of the three methods had the most obvious effect on improving the *mAP* of the model. Figure 8b shows that the *F*1 score of the model was higher than that of the other models when applying all three data augmentation methods. Table 3 shows the detailed results of the above five methods when used on the test set.

Table 3 shows that the *mAP* values of the YOLOv5-AT models were significantly improved when Mosaic, Mixup and SAM Copy-Paste were used compared to when no data augmentation method was used. For the *mAP*, the Mosaic method showed the most obvious improvement, about 5.4% on average, compared to the Mixup method (about 0.7%) and the SAM Copy-Paste method (about 1.3%). The *mAP* of the YOLOv5-AT model was improved by 7.3% on average compared to the No-Augment condition when the three data augmentation methods were combined. Thus, the application of data augmentation methods can make up for the low *mAP* values of object detection models that are caused by insufficient data. Figure 9 shows the actual detection effect of the different data augmentation methods.

### 3.3. Comparison Analysis of Different Dataset Sizes

Further, to prove that the YOLOv5-AT model combined with the data augmentation methods can solve the problem of the low accuracy of small datasets, based on the original training set, different proportions (20%, 30%, 50% and 70%) of the training set were used to retrain the YOLOv5-AT model. The results are shown in Table 4.

Table 4 shows that there was a positive correlation between the *mAP* of the model and the number of training sets. The *mAP* of the model continued to decrease as the number of training sets decreased. The *mAP* was reduced by only 2.3% when the number of samples was reduced by 50%. The *mAP* was only reduced by 1.6% when the number of samples was reduced by 30%. When the number of samples was only 20%, the *mAP* could still reach 72.8%. Moreover, the *mAP* of the model was improved by 17.5% under the condition of the same number of training sets by applying the data augmentation methods. It can be seen that YOLOv5-AT combined with the data augmentation methods can better solve the problem of the low *mAP* values of a small dataset.

### 3.4. Ablation Experiment

The effects on performance when removing or adding some features of the detection algorithm (Lawal et al. [39]) were observed. The attention mechanism introduced in the Conv-AT block was explored and studied in order to verify the performance of the YOLOv5-AT model (Chaudhari et al. [40]). The influence of the attention mechanism on the performance of the model introduced in the Conv-AT block is shown in Table 5.

Table 5 shows that the *mAP* of the model was 83.4% without the SA block and CA block. After introducing the CA block and SA block separately, the *mAP* of the model reached 84.0% and 84.2%: increases of 0.6% and 0.8%, respectively. These results proved that the feature information from the perspective of a spatial angle extracted by the SA block could improve the *mAP* efficiently compared to the feature information from the perspective of the channel extracted by the CA block. The *mAP*, *P*, *R* and *F*1 score of the model were 84.5%, 87.8%, 74.5% and 80.61%, respectively, which were 1.2%, 0.9%, 1.5% and 1.2% higher than those of the benchmark model after introducing the SA block and CA block at the same time. Thus, the combination of the CA block and SA block can reduce the influence of a complex environment and achieve higher *mAP* values for green orange detection under a background with a similar color.

### 3.5. Application of the YOLOv5-AT Model to Other Green Fruits

In order to verify the accuracy of the YOLOv5-AT model on other green fruits, green tomatoes and green persimmons were selected as the detection objects to train the YOLOv5-AT model. The results are shown in Table 6.

Table 6 shows that the performance of the same model on different detection objects was not the same, which was directly related to the difficulty in detecting objects and the image quality of the dataset. Compared to the green orange dataset, the YOLOv5-AT model performed better on the green tomato dataset, obtaining a *mAP* of 98% (298 images used for the model). This may be because the green tomato dataset was relatively simple, with fewer targets in a single image, a simple background and a low occlusion degree. Although the green persimmon dataset only had 198 images, the *mAP* of the YOLOv5-AT model reached 85.1%. Figure 10 shows the actual detection effect of the model on green tomatoes and green persimmons.

Figure 10 shows that the proposed method could accurately identify the target green fruit under a background with a similar color. On the one hand, this could be due to feature information being retained as much as possible throughout the concatenation operation. On the other hand, the YOLOv5-AT model reconstructs the feature information from the perspectives of space and channel by introducing the Conv-AT block and the CA and SA attention mechanisms. Therefore, the detection accuracy of the model was enhanced. However, the model may miss targets and perform false detection when the detection target is over-occluded, severely exposed and small.

## 4. Discussion

Due to green fruits possessing a similar color to the background of their images and the difficulty in collecting enough datasets, the existing detection models have all shown poor detection performance, affecting early orchard yield estimations and the allocation of water and fertilizer resources. Introducing the Conv-AT block and data augmentation methods into the YOLOv5 model enables important features to be extracted and transmitted from the input image and the original small dataset to be enlarged. The extracted features contain space and channel information, which can accelerate the convergence speed of the network and improve the detection accuracy. The application of data augmentation methods can increase the diversity of a dataset to provide a foundation for training detection models. By combining a Conv-AT block and data augmentation methods, green fruits can be detected even in cases of small datasets or images with similar background colors.

To do so, the most relevant factors affecting the results of detection models need to be analyzed. In this research, the influence of factors such as the angle of insolation, weather conditions and canopy density were taken into account during image acquisition. Consequently, images were collected under a variety of influencing factors. From the analysis results, the impact of the angle of insolation primarily manifested in exposure. When the angle of insolation is inappropriate, it can lead to improper exposure of the image, thereby degrading the image quality and leading to a decrease in detection accuracy. The effects of canopy density are obvious. When the canopy is luxuriant, it can cause partial or serious occlusion of the fruits, thereby reducing the accuracy of model detection. Therefore, we need to study the following aspects further in the future. Firstly, we will collect more images with varying canopy densities and shooting angles and, through the study of new algorithms, investigate ways to mitigate the impact of these factors in order to enhance the detection accuracy of the model. Secondly, when the dataset is limited and there are numerous influencing factors, transfer learning can be considered for model training to accelerate the training process. Finally, the channel pruning algorithm could be applied to reduce the size of the model, which could be considered for configuration in a mobile controller that is more appropriate for fieldwork. 

## 5. Conclusions

In summary, a YOLO object detection model based on the Conv-AT block, YOLOv5-AT, is proposed to achieve green fruit detection under a similar color background, and Mosaic, Mixup and SAM Copy-Paste are combined to expand the small dataset. We designed a CBR block and Conv-AT block on the basis of the original YOLOv5 model. The CBR block is used to extract local key information from the feature information. The Conv-AT block is used to obtain key information from different angles in the feature information. The experimental results from the green orange dataset show that the *mAP* of the YOLOv3-AT, YOLOv5-AT and YOLOv7-AT models reaches 84.3%, 84.6% and 79.1%, respectively. Compared to the benchmark model, it increased by 0.7%, 1.7% and 2.4%, respectively. The *P*, *R*, *F*1 score and *mAP* of the proposed YOLOv5-AT model on the test set are 86.3%, 76.3%, 81.0% and 84.6%, respectively, which are 0.3%, 3.4%, 2.1% and 1.7% higher than those of the benchmark model. At the same time, the Mosaic, Mixup and SAM Copy-Paste methods are used to expand the original data. The experimental results show that the YOLOv5-AT model combined with the three data augmentation methods can obtain a maximum improvement of 17.5% in the case of retaining different proportions of datasets. Therefore, the YOLOv5-AT model combined with data augmentation methods effectively achieves the accurate detection of green fruit targets in small datasets under a background with a similar color.

## Figures and Tables

**Figure 1 foods-13-01060-f001:**
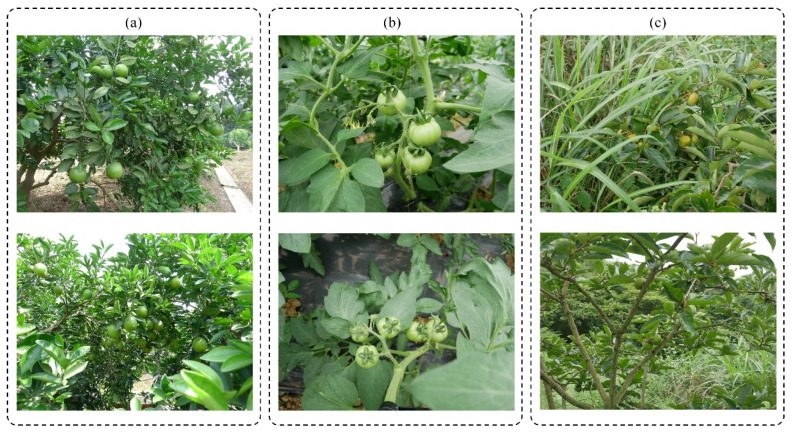
Sample images of different green fruits. (**a**) Green oranges; (**b**) green tomatoes; (**c**) green persimmons.

**Figure 2 foods-13-01060-f002:**
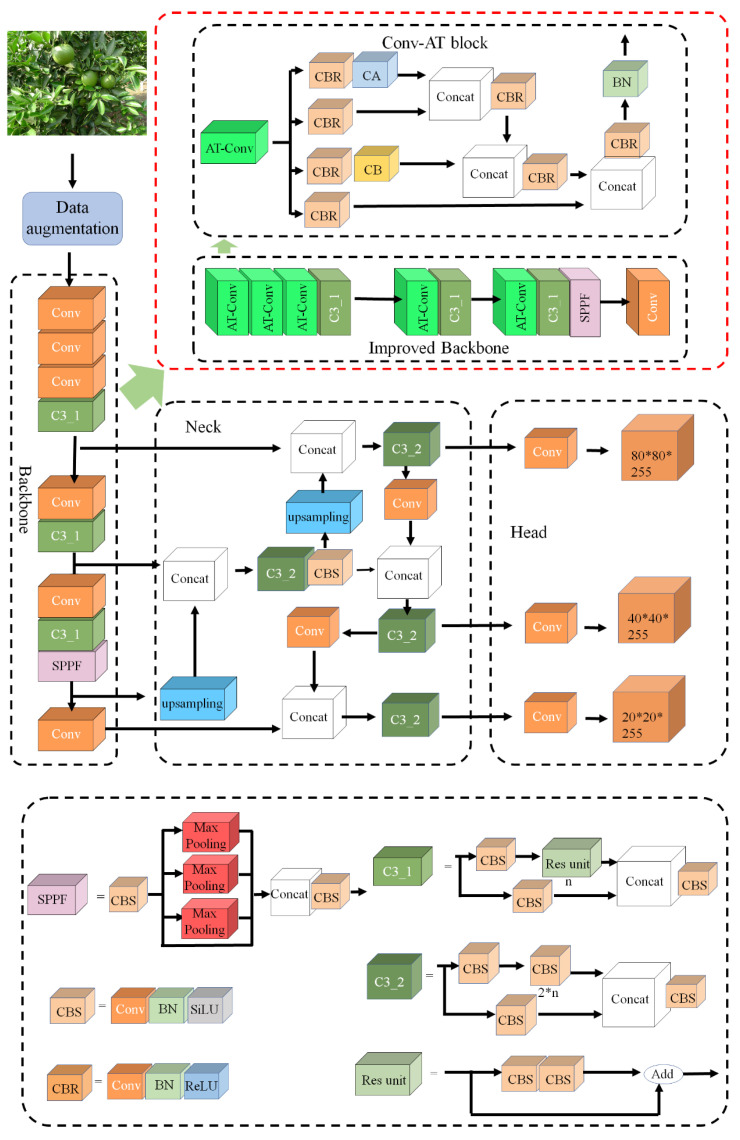
YOLOv5 and schematic diagram of its improved technique.

**Figure 3 foods-13-01060-f003:**
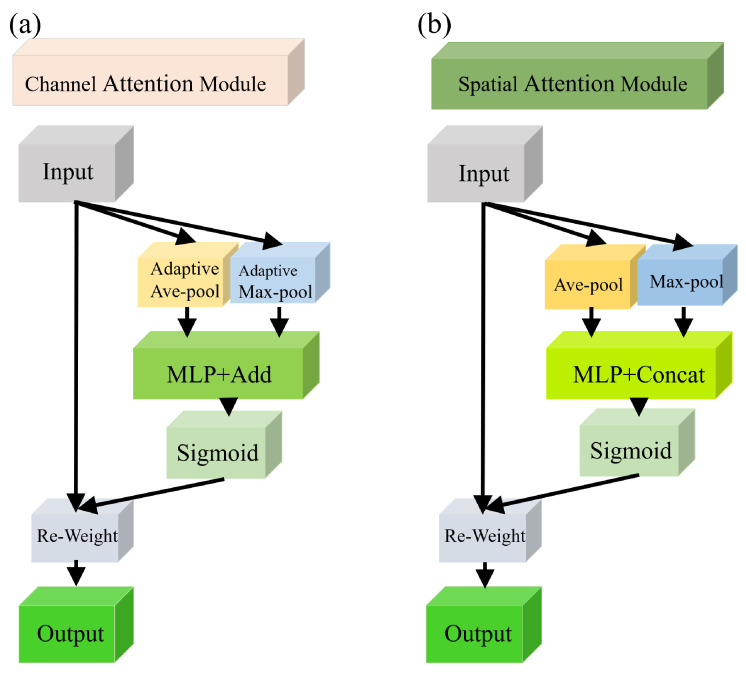
The structures of the CA and SA mechanisms. (**a**) CA mechanism. (**b**) SA mechanism.

**Figure 4 foods-13-01060-f004:**
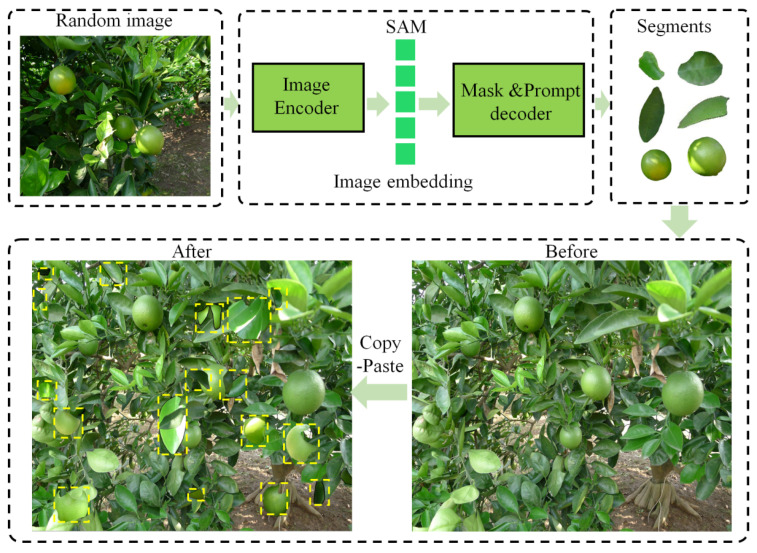
SAM Copy-Paste workflow diagram. Note: The yellow dotted frame in the picture represents the new leaves and oranges added to the original image by Copy-Paste.

**Figure 5 foods-13-01060-f005:**
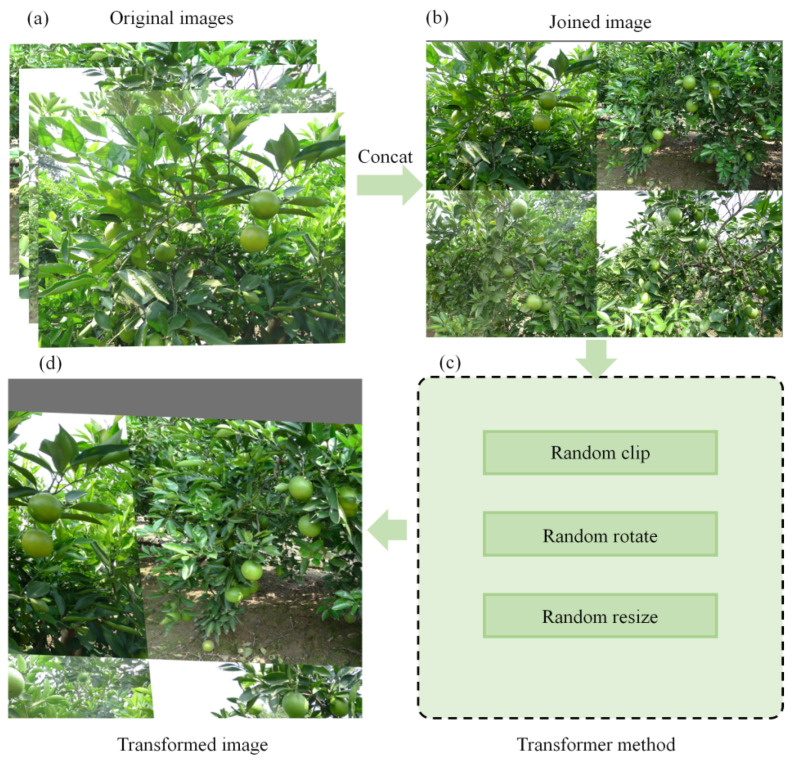
Mosaic workflow diagram. (**a**) Original image. (**b**) Joined image. (**c**) Transformer method. (**d**) Transformed image.

**Figure 6 foods-13-01060-f006:**
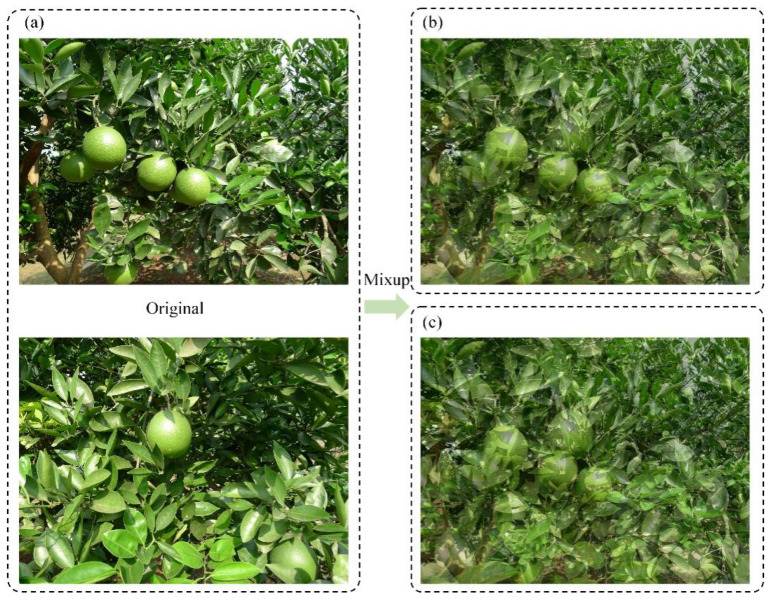
Mixup workflow diagram. (**a**) Original image; (**b**) the Mixup image when *λ* = 0.35; (**c**) the Mixup image when *λ* = 0.45.

**Figure 7 foods-13-01060-f007:**
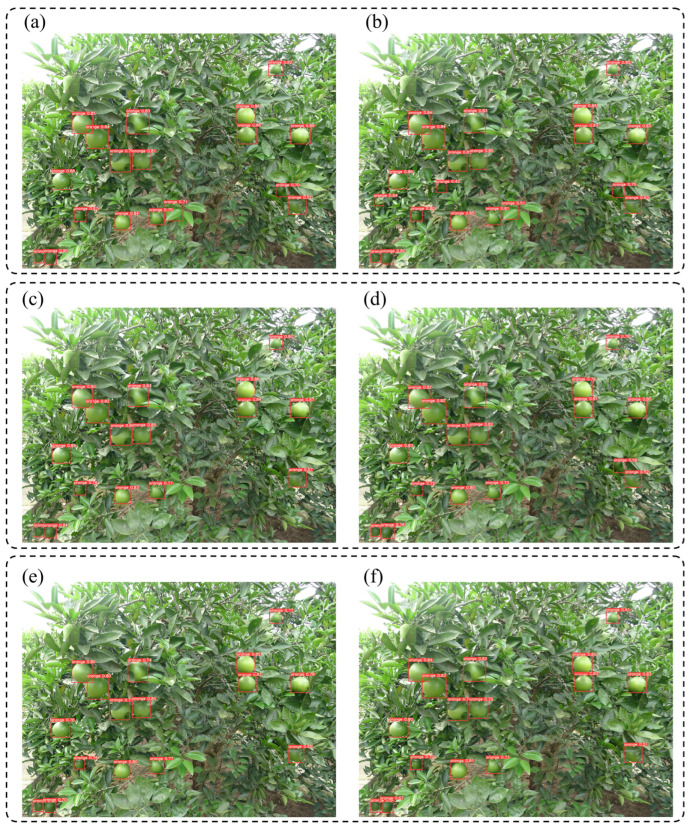
Comparison of the detection results of the YOLO models for green oranges. Note: The red boxes are detected targets. (**a**) YOLOv5. (**b**) YOLOv5-AT. (**c**) YOLOv3. (**d**) YOLOv3-AT. (**e**) YOLOv7. (**f**) YOLOv7-AT.

**Figure 8 foods-13-01060-f008:**
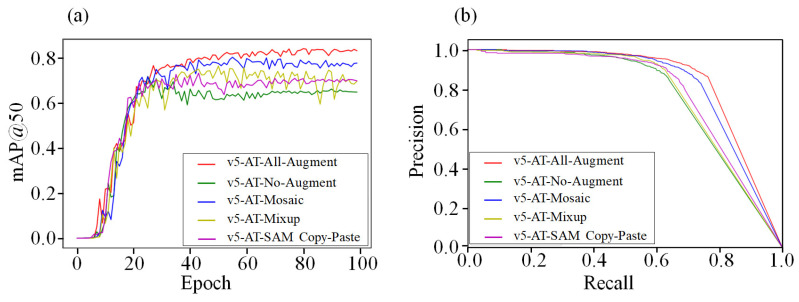
The results of the YOLOv5-AT model compared to different data augmentation methods. (**a**) The changes in the *mAP* values of the training process. (**b**) Precision recall curve.

**Figure 9 foods-13-01060-f009:**
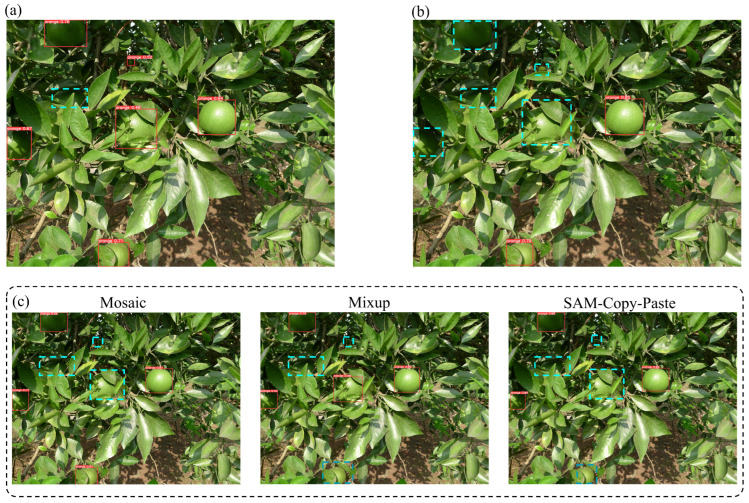
Actual detection effect of the different data augmentation methods. Note: The blue boxes are undetected targets, and the red boxes are detected targets. (**a**) All three data augmentation methods applied simultaneously. (**b**) No data augmentation method applied. (**c**) Three methods applied individually.

**Figure 10 foods-13-01060-f010:**
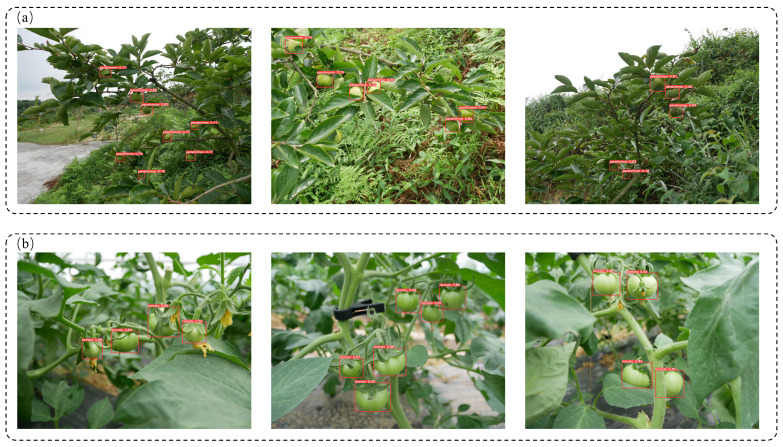
Actual detection effect of green tomatoes and green persimmons. (**a**) Green persimmons; (**b**) green tomatoes.

**Table 1 foods-13-01060-t001:** Composition of the datasets for the training and test.

Dataset	Resolution	Training Set	Test Set	Target Box
Green oranges	2560 × 1920	326	142	4417
Green persimmons		198	83	1212
Green tomatoes		219	94	751

**Table 2 foods-13-01060-t002:** Comparison results of different YOLO models.

Model	*P*	*R*	*F*1	*mAP*
YOLOv5-AT	86.3	76.3	81.0	84.6
YOLOv5	86	72.9	78.9	82.9
YOLOv3-AT	85.1	76.1	80.3	84.3
YOLOv3	83.8	75.2	79.3	83.6
YOLOv7-AT	81	74.5	77.6	79.1
YOLOv7	80.7	71.3	75.7	76.7

**Table 3 foods-13-01060-t003:** Test results of the YOLOv5-AT models with different augmentation methods.

	*P*	*R*	*F*1	*mAP*
All-Augment	86.3	76.3	81.0	84.6
Mosaic	85.2	72.5	78.3	82.7
Mixup	89.0	63.0	73.8	78.0
SAM Copy-Paste	85.5	66.7	74.9	78.6
No-Augment	84.9	63.9	72.9	77.3

**Table 4 foods-13-01060-t004:** YOLOv5-AT detection results with different proportions of the training set used. Note: ‘√’ indicates the use of data augmentation methods, while ‘×’ signifies not using data augmentation methods.

Proportion	Data Augmentation	*P*	*R*	*F*1	*mAP*
20%	√	81.0	63.5	71.2	72.8
	×	72.7	50.8	59.8	56.3
30%	√	84.2	68.8	75.7	78.0
	×	78.3	54.2	64.1	60.5
50%	√	86.6	72.7	79.0	82.2
	×	85.1	56.5	67.9	65.5
70%	√	86.3	74.8	80.1	82.9
	×	79.6	59.3	68.0	68.2
100%	√	86.3	76.3	81.0	84.6
	×	84.9	63.9	72.9	77.3

**Table 5 foods-13-01060-t005:** Ablation experiment results. Note: ‘√’ indicates the utilization of the corresponding attention mechanism module, whereas ‘×’ signifies the non-utilization of the respective attention mechanism module.

CA	SA	*P*	*R*	*F*1	*mAP*
×	×	85.4	74.8	79.8	83.4
√	×	86	74.9	80.1	84.0
×	√	85.6	75.4	80.1	84.2
√	√	86.3	76.3	81.0	84.6

**Table 6 foods-13-01060-t006:** Performance of the YOLOv5-AT model on other green fruits.

Fruit Species	*P*	*R*	*F*1	*mAP*
Green tomatoes	95.1	93.4	94.2	98.0
Green persimmons	85.1	76.0	80.3	85.1

## Data Availability

The original contributions presented in the study are included in the article/Appendix A, further inquiries can be directed to the corresponding author.

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
