# Peer review of "Green Fruit Detection with a Small Dataset under a Similar Color Background Based on the Improved YOLOv5-AT"

_foods, 2024, doi:10.3390/foods13071060_

Round 1

Reviewer 1 Report

Comments and Suggestions for Authors

The study addresses the challenge of accurately detecting green fruits in small datasets, crucial for estimating orchard yield and resource allocation. By proposing an enhanced YOLOv5 model with Conv-AT block, SA, CA attention blocks, and data augmentation, the model achieved higher mAPs for green orange, green tomato, and green persimmon compared to other YOLO models, demonstrating effectiveness in overcoming challenges associated with limited dataset size and complex backgrounds. The idea is good but there are some points that should be considered before publication:

- The similarity index, 31%, is high a little bit. Please try to reduce this.

- Please don’t reuse the words that you already used in title for keywords.

- lines 16 1120: Please don’t use active voice sentence. Recheck all throughout the article.

-Table 1: How can you divide figures 70%,30%? Give details at this point please. Additionally train and test part is ok but what about validation part?

- Equations should be written using word equation function instead of saving as a figure. Their qualities are not good.

- It is crucial to elaborate on the approach used to evaluate the prediction results. How was the accuracy calculated? When dealing with multiple classes, it is advisable to present a comprehensive analysis incorporating metrics such as Average Accuracy, Error Rate, Precision, Recall, and F-score, 

Comments on the Quality of English Language

Moderate editing of English language required.

Author Response

- Reviewer 1:

1. The similarity index, 31%, is high a little bit. Please try to reduce this.

Reply: Thank you for the suggestion. We have carefully reviewed the literature and made further adjustments to reduce the similarity index. The index has now been lowered to a more appropriate level. We hope that these modifications meet your expectations.

  1. Please don’t reuse the words that you already used in title for keywords.

Reply: Thank you for the suggestion. We have ensured that the keywords do not directly replicate the words used in the title to avoid redundancy and enhance the clarity and specificity of the keyword selection. In the revised manuscript, we have changed the keywords to “green fruit; precision detection; data augmentation methods; CA and SA attention block; feature information”.

  1. lines 16 1120: Please don’t use active voice sentence. Recheck all throughout the article.

Reply: Thank you for the suggestion. We have reviewed the entire article to ensure consistent use of the passive voice and have made any necessary adjustments to comply with your recommendation.

  1. Table 1: How can you divide figures 70%,30%? Give details at this point please. Additionally train and test part is ok but what about validation part?

Reply: Thank you for the suggestion. In this research, the green orange dataset initially consisted of 468 images. These figures were subsequently divided into training and test sets using a 7:3 ratio. The original dataset, consisting of 468 images, was randomly split: 326 images (70% of the total, calculated as 468 * 0.7) were selected for the training set, while the remaining 142 images (30% of the total, calculated as 468 * 0.3) were designated as the test set. The division of the green tomato dataset and the green persimmon dataset followed the same procedure as that of the green orange dataset.

Usually, the dataset could be divided into training set, validation set and test set. The training set is utilized for training the model, where input data and their corresponding correct outputs are fed into the algorithm. Through this process, the model learns from the data in order to make accurate predictions. The validation set is often used to fine-tune the model hyperparameters and avoid overfitting. The test set is used to evaluate the performance of the trained model. Nevertheless, due to the limited number of images in our dataset, following suggestions from other research studies [1-3] which caution against using a smaller test set that could potentially misrepresent the model's capabilities, we decided not to partition a separate validation set in this investigation.

References:

[1] Ma Jie, Ange Lu, Chen Chen, Xiandong Ma, and Qiucheng Ma. "Yolov5-Lotus an Efficient Object Detection Method for Lotus Seedpod in a Natural Environment." Computers and Electronics in Agriculture 206 (2023): 107635.

[2] Tang Yuanchao, Zhou Hao, Wang Hongjun, and Zhang Yunqi. "Fruit detection and positioning technology for a Camellia oleifera C. Abel orchard based on improved YOLOv4-tiny model and binocular stereo vision." Expert systems with applications 211 (2023): 118573.

[3] Sun Qixin, Chai Xiujuan, Zeng Zhikang, Zhou Guomin, Sun Tan, "Noise-tolerant RGB-D feature fusion network for outdoor fruit detection." Computers and Electronics in Agriculture 198 (2022): 107034.

  1. Equations should be written using word equation function instead of saving as a figure. Their qualities are not good.

Reply: Thank you for the suggestion. We have revised the equations in the revised manuscript using the word equation function instead of saving them as figures to improve their quality.

  1. It is crucial to elaborate on the approach used to evaluate the prediction results. How was the accuracy calculated? When dealing with multiple classes, it is advisable to present a comprehensive analysis incorporating metrics such as Average Accuracy, Error Rate, Precision, Recall, and F-score, 

Reply: Thank you for the suggestion. In this study, the precision (P), recall (R), average precision (AP) and score (F1) are used as evaluation indicators.

where TP denotes the number of positive samples predicted as positive, FN denotes the number of negative samples predicted as negative, FP denotes the number of negative samples predicted as positive and n is the number of categories. The intersection set IoU indicates the overlap ratio between the predicted bounding box and true bounding box. In this research, the threshold for the IoU was set as 0.5. The sample was defined as a TP when the IoU was greater than 0.5. If the IoU was less than 0.5, this sample was defined as a false sample or FP. In this research, the model was trained independently on each of the three green fruit datasets: the green orange dataset, the green tomato dataset, and the green persimmon dataset, with n being 1 in all cases, indicating that the model dealt with each single class separately. When dealing with single-class or multiple-classes, the mAP and F1 can serve as comprehensive analysis metrics to reflect the performance of the model.

Reviewer 2 Report

Comments and Suggestions for Authors

The manuscript presents a study on the detection of green fruit in the field using digital images and deep learning. This type of problem still lacks robust solutions, so new research on the subject is welcome. However, the manuscript has a number of problems that need to be properly addressed, as detailed below.
- Title: the term “near color background” does not make much sense. I suggest replacing “near” with “similar”.
- The “Materials and Methods” section needs more detail. A large part of the section is dedicated to describing the neural networks’ architecture, but there is not enough information, for example, about the image augmentation process: How many new images were generated using each technique? How similar/dissimilar are the new images compared with the original ones? Was augmentation applied after the division into training and test sets (correct procedure), or before the division (wrong procedure)? The authors should provide enough information for reproduction.
- Figure and Table captions: these should be as self-contained as possible, but many of the captions found throughout the manuscript are not nearly descriptive enough.
- There is no discussion about the results reported in the manuscript. There are many factors that can influence the detection of fruits in the field, including weather, angle of insolation, color of leaves, fruit maturity, canopy density, among others. The authors should discuss at least the most relevant of those factors, explaining how the results were affected when those factors changed. This is important for the reader to understand how the models operate, and to identify the conditions under which the models do not perform as well as expected, thus guiding future research efforts.
- Talking about future research, there is no mention about the next steps into the development of a truly robust technology for fruit detection. The proposed method has some evident weaknesses that should be thoroughly discussed in the article, and then the authors should suggest some possible avenues to close those research gaps and improve the overall performance.

Comments on the Quality of English Language

The language, although mostly intelligible, is often clunky and hard to read. A thorough revision by someone more familiar with the English language is recommended.

Author Response

- Reviewer 2:

The manuscript presents a study on the detection of green fruit in the field using digital images and deep learning. This type of problem still lacks robust solutions, so new research on the subject is welcome. However, the manuscript has a number of problems that need to be properly addressed, as detailed below.
1. Title: the term “near color background” does not make much sense. I suggest replacing “near” with “similar”.

Reply: Thank you for your suggestion regarding the title. We have enhanced clarity and accuracy by replacing 'near' with 'similar' throughout the revised manuscript, ensuring that the related correction has been consistently applied.

  1. The “Materials and Methods” section needs more detail. A large part of the section is dedicated to describing the neural networks’ architecture, but there is not enough information, for example, about the image augmentation process: How many new images were generated using each technique? How similar/dissimilar are the new images compared with the original ones? Was augmentation applied after the division into training and test sets (correct procedure), or before the division (wrong procedure)? The authors should provide enough information for reproduction.

Reply: Thank you for the suggestion. We have added a detailed description in the part of materials and methods in the revised manuscript. The responses to the above questions are as follows:

1)How many new images were generated using each technique?

Reply: In this research, the SAM Copy-Paste, Mosaic and Mixup were used to enhance the dataset according with different probability and parameters. Mosaic was applied with a probability of 0.8. Mixup was applied with a probability of 0.3. SAM Copy-Paste was applied with a probability of 1 and pasted 4 instances of green oranges and 15 instances of leaves into each image.

2)How similar/dissimilar are the new images compared with the original ones?

Reply: Different data augmentation methods would have different influence on original image.

The application of the SAM Copy-Paste technique enables pasting leaves and green fruits from other images onto the original image, thereby simulating a variety of complex conditions for green fruits in real-world scenarios and enhancing the diversity within the dataset. The pasted elements are indicated by yellow boxes, and their actual effect can be seen in Figure 1.

Figure 1 Comparison of image before and after SAM-Copy-Paste processing

The Mosaic application fuses four images together to enhance the diversity of the original dataset. The resulting image incorporates partial features from each of the original images. The actual effect of this fusion can be observed in Figure 2.

Figure 2 Comparison of image before and after Mosaic processing

The application of Mixup generates a new image by using linear interpolation between two original images, which is mainly employed to improve the model's performance in detecting complex environments to a certain extent. The actual effect can be observed in Figure 3.

Figure 3 Comparison of image before and after Mixup processing

3)Was augmentation applied after the division into training and test sets (correct procedure), or before the division (wrong procedure)?

Reply: Thank you for your suggestion. The data augmentation method was applied after the division into training and test set.
3. Figure and Table captions: these should be as self-contained as possible, but many of the captions found throughout the manuscript are not nearly descriptive enough.

Reply: Thank you for your suggestion regarding the figure and table captions. We have revised the captions throughout in revised manuscript to provide more detailed descriptions that stand on their own.
4. There is no discussion about the results reported in the manuscript. There are many factors that can influence the detection of fruits in the field, including weather, angle of insolation, color of leaves, fruit maturity, canopy density, among others. The authors should discuss at least the most relevant of those factors, explaining how the results were affected when those factors changed. This is important for the reader to understand how the models operate, and to identify the conditions under which the models do not perform as well as expected, thus guiding future research efforts.

Reply: Thank you for the suggestion. The detail information about the collection of images was added to the Part 2.1 (Acquisition of images). And the discussion about the most relevant of above factors (canopy density, weather and so on) to explain how the results were affected is added to the Part 4 (Discussion) in the revised manuscript. The detail content was as follows:

The most relevant factors affecting the results of detection models need to be analyzed. In this research, the influence of factors such as angle of insolation, weather conditions, and canopy density were taken into account during image acquisition. Consequently, images were collected under a variety of influencing factors. From the analysis results, the impact of angle of insolation primarily manifested in exposure. When the angle of insolation is inappropriate, it can lead to improper exposure of the image, thereby degrading image quality and leading to a decrease in detection accuracy. The effects of canopy density are obvious. When the canopy is luxuriant, it can cause partial or serious occlusion of the fruits, thereby reducing the accuracy of model detection. Therefore, we need to study the following aspects further in the future. Firstly, we will collect more im-ages with varying canopy densities and shooting angles, and through the study of new algorithms, investigate ways to mitigate the impact of these factors in order to enhance the detection accuracy of the model. Secondly, when the dataset is limited and there are numerous influencing factors, transfer learning can be considered for model training to accelerate the training process. Finally, the channel pruning algorithm could be applied to reduce the size of the model, which could be considered for configuration in a mobile con-troller that is more appropriate for field work.

  1. Talking about future research, there is no mention about the next steps into the development of a truly robust technology for fruit detection. The proposed method has some evident weaknesses that should be thoroughly discussed in the article, and then the authors should suggest some possible avenues to close those research gaps and improve the overall performance.

Reply: Thank you for the suggestion. A discussion was added in Part 4 (Discussion) of the revised manuscript to more effectively elucidate the model's shortcomings and outline future directions for improvement. The detail content was as follows:

Due to green fruits possessing a similar color to the background of their images and the difficulty of collecting enough datasets, the existing detection models have all shown poor detection performance, affecting early orchard yield estimation and the allocation of water and fertilizer resources. Introducing the Conv-AT block and data augmentation methods into the YOLOv5 model enables important features to be extracted and transmit-ted from the input image and the original small dataset to be enlarged. The extracted feature contains space and channel information, which can accelerate the convergence speed of the network and improve detection accuracy. The application of data augmentation methods can increase the diversity of a dataset to provide a foundation for training detection models. By combining a Conv-AT block and data augmentation methods, green fruit can be detected even in cases of small datasets or images with similar background colors.

To do so, the most relevant factors affecting the results of detection models need to be analyzed. In this research, the influence of factors such as angle of insolation, weather conditions, and canopy density were taken into account during image acquisition. Consequently, images were collected under a variety of influencing factors. From the analysis results, the impact of angle of insolation primarily manifested in exposure. When the angle of insolation is inappropriate, it can lead to improper exposure of the image, thereby degrading image quality and leading to a decrease in detection accuracy. The effects of canopy density are obvious. When the canopy is luxuriant, it can cause partial or serious occlusion of the fruits, thereby reducing the accuracy of model detection. Therefore, we need to study the following aspects further in the future. Firstly, we will collect more im-ages with varying canopy densities and shooting angles, and through the study of new algorithms, investigate ways to mitigate the impact of these factors in order to enhance the detection accuracy of the model. Secondly, when the dataset is limited and there are numerous influencing factors, transfer learning can be considered for model training to accelerate the training process. Finally, the channel pruning algorithm could be applied to reduce the size of the model, which could be considered for configuration in a mobile con-troller that is more appropriate for field work.

Round 2

Reviewer 1 Report

Comments and Suggestions for Authors

From my side, it  can be published as is.

Comments on the Quality of English Language

Minor editing of English language required.

Reviewer 2 Report

Comments and Suggestions for Authors

The manucript has improved considerably and I am satisfied with the changes introduced by the authors.

Comments on the Quality of English Language

There are still some minor language problems, but overall the quality of the writing has improved significantly.